Brief Communication

# In vivo imaging of the human brain with the Iseult 11.7-T MRI scanner

Nicolas Boulant [1], Franck Mauconduit [1], Vincent Gras[1], Alexis Amadon [1], Caroline Le Ster[1], Michel Luong[2], Aurélien Massire[3], Christophe Pallier[4], Laure Sabatier [5], Michel Bottlaender[1], Alexandre Vignaud[1] & Denis Le Bihan [1] ✉

The understanding of the human brain is one of the main scientific challenges of the twenty-first century. In the early 2000s, the French Atomic Energy Commission launched a program to conceive and build a human magnetic resonance imaging scanner operating at 11.7 T. We have now acquired human brain images in vivo at such a magnetic field. We deployed parallel transmission tools to mitigate the radiofrequency field inhomogeneity problem and tame the specific absorption rate. The safety of human imaging at such high field strength was demonstrated using physiological, vestibular, behavioral and genotoxicity measurements on the imaged volunteers. Our technology yields $T_2$ and $T_2^*$-weighted images reaching mesoscale resolutions within short acquisition times and with a high signal and contrast-to-noise ratio.

Understanding the human brain, normal or diseased, is one of the most substantial scientific challenges of the twenty-first century, with academic, medical, societal and economic implications. In this context, highly funded international research programs are building on neuroimaging, and especially magnetic resonance imaging (MRI), as an essential approach to obtain in situ and noninvasive maps of brain anatomy, function and structural connectivity in patients and healthy volunteers[1]. Continuing efforts aim at boosting functional[2] and diffusion[3,4] MRI to explore the human brain in vivo at finer spatial resolutions[5].

Benefiting from a supralinear gain in sensitivity[6,7], this goal can be reached by boosting the magnetic field, B0, at which MRI systems operate. The Iseult project, launched in 2001, was based on this vision and aimed at building an MRI magnet operating at 11.7 T to investigate the human brain at a mesoscopic scale. After nearly 20 years of research and development mainly led by the Department of Astrophysics, Particle Physics, Nuclear Physics and Associated Instrumentation of the French Atomic Energy Commission (CEA), the 132-ton 11.7-T magnet (Fig. 1), together with the scanner equipment, in 2021 delivered images of the human brain ex vivo[8]. The field homogeneity and temporal stability constraints for MRI were met thanks to the careful design of the magnet[8,9].

When increasing B0, the radiofrequency (RF) field used to excite the nuclear spins becomes more inhomogeneous, a long-standing and major problem in MRI due to the shortening of the wavelength of the RF waves used (500 MHz at 11.7 T for the hydrogen nucleus of water), leading to severe artifactual signal variations in the images visible already on the ex vivo brain[8]. We tackled this problem by deploying parallel transmission hardware and software, first developed by the laboratory at 7 T and further optimized for 11.7 T. The specific absorption rate (SAR), also growing with B0 (ref. 10), was tamed with virtual observation points[11] and dedicated RF pulse design algorithms[12].

Approval from the regulatory body and ethics committee to perform initial tests in vivo on 20 healthy adult participants was granted early in 2023 to validate the general 11.7-T MRI set-up and safety. Until now, the highest magnetic field used for MRI on humans was 10.5 T, at the University of Minnesota in the United States[13]. Overall, for small flip angle excitations and large flip angle refocusing pulses, normalized root mean square errors (n.r.m.s.e.) equal or lower than 13% thereby could be obtained at 11.7 T over the whole brain, the latter value being the intrinsic inhomogeneity achieved at 3 T with volume coils[14]. Inversion pulses remained difficult to achieve given the architecture of the coil, leaving very localized artifacts despite reaching 8% n.r.m.s.e. Static

[1]NeuroSpin, CEA, CNRS, Paris-Saclay University, Gif-sur-Yvette, France. [2]DACM, CEA, Paris-Saclay University, Gif-sur-Yvette, France. [3]Siemens Healthcare SAS, Courbevoie, France. [4]Cognitive Neuroimaging Unit, NeuroSpin, INSERM, CEA, CNRS, Paris-Saclay University, Gif-sur-Yvette, France. [5]DIREI, CEA, Paris-Saclay University, Gif-sur-Yvette, France. ✉e-mail: denis.lebihan@cea.fr

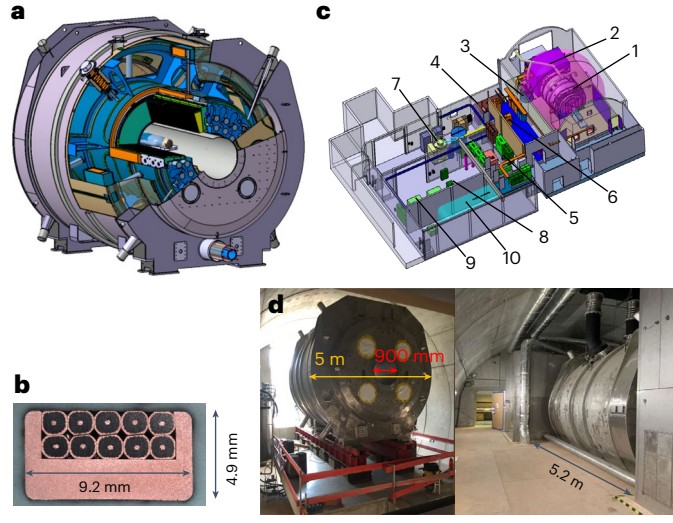

**Fig. 1 | Iseult magnet and installation set-up. a**, Schematic of the magnet showing the main coil and the two actively shielding coils. The magnet weighs 132 tons, has an outer diameter and total length of just under 5 m and the bore diameter is 90 cm. **b**, Cross-section of the wire used for the magnet showing the packing of the ten strands of NbTi-Cu wire within a copper channel. **c**, General view of the Iseult MRI inside the NeuroSpin building: (1) magnet (2), Faraday cages, (3) satellite, (4) dump resistor, (5) magnet control command and power convertor, (6) Siemens Healthineers MRI equipment, (7) helium liquefier, (8) helium pumps, (9) compressors and (10) helium balloon. **d**, Left: initial installation of the magnet within a dedicated arch (10 m in diameter) at NeuroSpin. Right: current side view of the magnet within the arch, extending above the ceiling and under the floor of the patient side room, going all the way in the back of the arch.

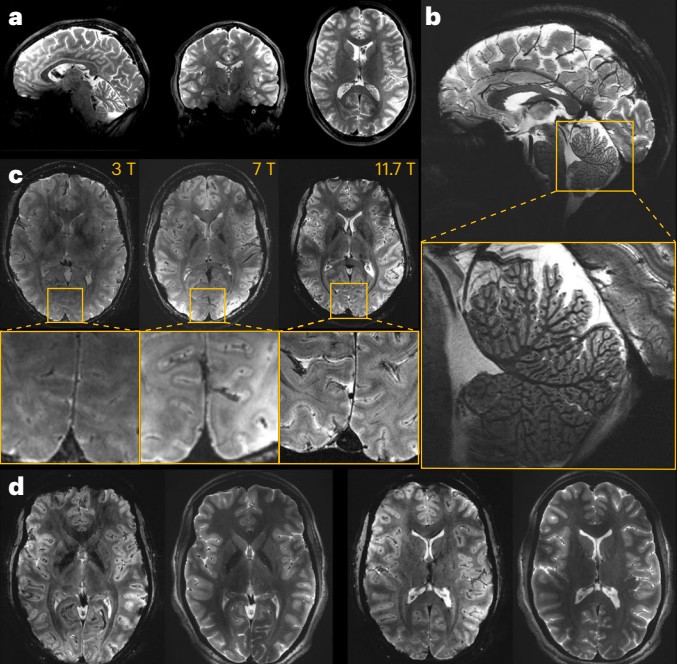

**Fig. 2 | In vivo images of the human brain obtained at 11.7 T. a**, In vivo 3D $T_2$ variable flip angle turbo spin-echo acquisition at 11.7 T with parallel transmission GRAPE universal pulses (resolution $0.55 \times 0.55 \times 0.55$ mm$^3$, acquisition time 13 min) demonstrating the high $B_1^+$ (RF) field homogeneity reached throughout the whole brain volume. **b**, In vivo $T_2^*$-weighted 2D GRE sagittal slice (resolution $0.2 \times 0.2 \times 1$ mm$^3$, acquisition time 8 min 30 s). **c**, $T_2^*$ weighted 2D GRE axial images acquired at 3 T (left), 7 T (middle) and 11.7 T (right) with identical acquisition times (4 min 17 s), while keeping similar contrast-to-noise ratio through adjusted acquisition parameters (FA (°), TR (ms) and TE (ms) of 27, 750 and 45; 34, 950 and 25; 27, 600 and 20 at 3, 7 and 11.7 T, respectively) and spatial resolution (0.5, 0.325 and 0.2-mm in-plane resolution for 3, 7 and 11.7 T, respectively, 1-mm thickness). **d**, The 11.7 T $T_2^*$-weighted 2D GRE axial images (resolution $0.19 \times 0.19 \times 1$ mm$^3$, acquisition time 5 min 16 s) juxtaposed to turbo spin-echo $T_2$-weighted images (resolution $0.3 \times 0.3 \times 1$ mm$^3$, acquisition time 4 min 26 s).

B0 shimming was performed up to second order for each participant, using a brain mask and a quadratic programming approach yielding on average 82.7-Hz standard deviation over the brain (0.17 ppm). After B0 and RF field mapping for each participant, following the RF pulse designs $T_2$ and $T_2^*$-weighted acquisitions were performed by deploying parallel transmission techniques (Fig. 2). The presented in vivo human brain images acquired at 11.7 T reveal good signal homogeneity consistent with the simulations, without severe RF field inhomogeneity artifacts. A 2D $T_2^*$-weighted acquisition was repeated at 3 and 7 T on different participants to reach similar contrast-to-noise ratio at the three field strengths, while keeping similar acquisition times, showing the gain in spatial resolution when increasing the field (Fig. 2c) and details around the calcarine fissure and within the cerebellum. Details within the cortical ribbon become clearly visible at 11.7 T and not at lower field strength due to poorer spatial resolution. Another approach was to keep the same protocol parameters (resolution, bandwidth, acquisition time, number of receive elements and so on) to visualize the signal to noise ratio (SNR) gain brought by field strength (Extended Data Fig. 1).

Whether human beings could withstand 1.5-hour scans at 11.7 T was uncharted territory, while vestibular effects have been reported in mice[15], as well as mild genotoxic effects with repeated or chronic field exposures[16]. This study therefore proceeded cautiously to gather evidence for the absence of harmful effects in more realistic conditions[17,18]. Physiologic, vestibular[19] and cognitive[20,21] measurements were implemented in the protocol to verify the innocuousness of the large magnetic field on humans. Those tests were performed on the 20 participants scanned at 11.7 T and on another cohort of 20 participants with the magnetic field switched off to disentangle any effect from a possible psychological bias (nocebo effect). The participants of the 0-T control group were not informed of the absence of the magnetic field and received the same instructions as the volunteers scanned at 11.7 T. Due to the absence of vibrations at 0 T, we mimicked the sound of MRI sequences with recordings played by a loudspeaker hidden at the back

of the magnet. Genotoxicity was also assessed but only for the 11.7-T group with blood samples drawn before and after the 90 min exposure to perform intra-participant comparisons. Across all the tests carried out in this protocol, the statistical analysis revealed no significant differences related to field exposure (analysis of variance, $P = 0.54$ for group and run effects, Extended Data Fig. 2 and Extended Data Table 1).

The present study, as granted by our regulatory body and ethics committee, was exploratory, requiring iterations of our pulse design methods and parameter optimization over the 20 participants to mitigate RF field inhomogeneity and converge toward the presented results, hence preventing us at this stage from assessing the reproducibility and exploring multiple contrasts besides (anatomical) $T_2$ and $T_2^*$. Furthermore, about two-thirds of the high-resolution scans were corrupted by motion. With the limited number of participants authorized in this initial protocol, we thereby could obtain good quality images with resolutions up to $0.19 \times 0.19 \times 1$ mm$^3$ resolution in about 5 min. However, those images indicate sufficient SNR to boost the spatial resolution further[22]. The next steps will thus focus on development and implementation of motion correction tools[23,24] together with highly accelerated sequences[25,26]. The deployment of more efficient RF coils[27], more receive channels count[5,28] and more powerful gradients[5] is also underway to further increase performance, especially to perform high-resolution functional MRI and diffusion tensor imaging or tractography, which was not possible within the scope of this preliminary study.

The safety and imaging data gathered in this first study confirm the applicability of MRI at 11.7 T; humans can safely tolerate such intense magnetic fields while imaging at mesoscale resolutions can be achieved on the brain in reasonable time. It brings to the fingertips of the neuroscience and medical community an opportunity to explore the brain in more detail. The higher resolution and the contrasts that ultrahigh-frequency MRI provides will certainly open a window of opportunity to better understand certain neurological conditions, and develop new disease biomarkers or therapeutic means. Among identified targets[29] are drug-resistant focal epilepsy (focal cortical dysplasia, mesial temporal sclerosis), multiple sclerosis (cortical lesions, veins), assessment of microvessels (brain tumors and angiogenesis, small metastases, chronic stroke), hippocampus status (Alzheimer's disease), the brain's basal ganglia (iron content, Parkinson's disease, deep brain stimulation) and imaging of nonproton nuclei (lithium in bipolar disorders, neurotransmitters, $^{23}$Na, $^{31}$P in energy metabolism). By revealing at a mesoscopic scale the structural–functional links that compose the human brain (cytoarchitectonic of neuron clusters and their connections), one could bridge the gap toward both the extensive body of knowledge we already have at the microscopic scale on circuits, neurons and synapses in animal models, and the human brain connectome at a macroscopic scale. This will allow us to confirm or refute our current hypotheses regarding its functioning and to generate new ones, and to investigate the biological mechanisms underlying our mental life, consciousness and better identify the mechanisms at work in mental illnesses (depression, autism, schizophrenia)[30].

## Online content

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

## Methods

### Data acquisition and processing

**Magnet.** The 132-ton 11.7-T magnet (Fig. 1) consists of a main superconducting coil made of a stack of 170 double pancakes coupled with a passive iron shim and an external power supply[9]. The field homogeneity currently is 0.9 ppm peak-to-peak over a 22-cm-diameter sphere with a 3-ppb per hour field drift at thermal equilibrium[8]. To operate the 182-km wire in superconductive, cryostable mode and provide a large safety buffer for MRI operation, the magnet is cooled at a temperature of 1.8 K (standard MRI magnets operate at 4.2 K) with a bath of 7,000 l of superfluid helium continuously maintained by a dedicated cryogenic plant located beneath the MRI scanner. The magnet is connected to its power supply and the dedicated cryogenic is installed underneath through a cryoline (Fig. 1).

**MRI.** The magnetic resonance sessions were 90 min long and targeted anatomical $T_1$, $T_2$ and $T_2^*$ contrasts. A home-made parallel transmission RF coil (pTx) with dedicated virtual observation points for SAR monitoring was used[31]. The coefficient of variation (standard deviation over mean) of the $B_1^+$ field in static RF shimming mode was 45% over the whole brain and on average over the participants. Increased power losses at 500 MHz combined with the pairing strategy of the 16 transmit elements to be fed by eight RF power amplifiers made inversion pulses particularly difficult to design, despite reaching 8% n.r.m.s.e. on average over the participants (5% min, 12% max). This underlined the necessity to have high power RF amplifiers (here 2 kW per channel) and the importance of low cable losses, transmitting efficiency, a high transmit channel count and optimizing channel pairing. The small flip angle excitation (n.r.m.s.e. roughly 8% over the 3D brain) and high flip angle refocusing (roughly 13%) pulses were designed with the $k_T$ point[32] and the gradient ascent pulse engineering algorithm (GRAPE)[33] approaches, respectively, using an active-set algorithm[12] with simultaneous $k$-space optimization[34] and under explicit hardware (peak power, average power, maximum gradient slew rate) and SAR constraints. The GRAPE pulse was a universal RF pulse[33] and was designed offline on a database constituted of the first six participant field maps. The RF and $\Delta B0$ field maps were acquired for each participant for the pulse design and shimming using 2D interferometric turbo-FLASH[35] at 5-mm isotropic resolution and 3D multi-echo (echo time (TE) 1.6, 3.5 and 6 ms) gradient echo sequences (GRE) acquisitions at 2.5 mm isotropic resolution respectively. Sequence parameters for the $T_2$-weighted variable flip angle turbo spin-echo acquisition were: resolution $0.55 \times 0.55 \times 0.55$ mm$^3$, repetition time (TR) 6 s, time of acquisition (TA) 13 min, the generalized autocalibrating partially parallel acquisitions framework (GRAPPA) $3 \times 2$, TE = 301 ms, matrix $400 \times 400 \times 320$ and bandwidth of 250 Hz per pixel. All acquisitions were performed with up to second-order shimming using quadratic programming and computed brain masks. Comparisons obtained with similar data acquired at 3 and 7 T revealed less than linear B0 field dispersions with field strength, indicating more than satisfying results in terms of effective B0 homogeneity at 11.7 T in the brain.

Slice-selective 'spoke' pTx pulses[36] were designed for the $T_2^*$-weighted 2D GRE acquisitions. The sequence parameters were flip angle (FA) 27°, TR = 600 ms, GRAPPA = 2, TE = 20 ms, resolution of $0.2 \times 0.2 \times 1$ mm$^3$, readout bandwidth of 40 Hz per pixel, TA = 4 min 17 s (two spokes[37] for the axial acquisition) or TA = 8 min 30 s (two-mode optimization time interleaved acquisition of modes[38] for the sagittal acquisition). Another $T_2^*$-weighted 2D GRE acquisition with a resolution of $0.19 \times 0.19 \times 1$ mm$^3$ (two spokes, TA = 5 min 16 s, same parameters as above otherwise) and a turbo spin-echo sequence of resolution of $0.3 \times 0.3 \times 1$ mm$^3$ (two-mode optimization time interleaved acquisition of modes, TR = 7 s, TE = 48 ms, bandwidth of 130 Hz per pixel, echo train length 9, GRAPPA = 4, TA = 4 min 26 s, seven slices) were implemented on the last volunteer. Careful analysis of the sequences aided with field monitoring[39] and sequence adjustments were performed

before the in vivo experiments to minimize field perturbations during the acquisitions[40]. For the $0.2 \times 0.2 \times 1$ mm$^3$ high-resolution 2D GRE experiments, similar scans were performed at 3 T and 7 T (FA = 27°, TR = 600 ms, GRAPPA = 2, TE = 20 ms, readout bandwidth of 40 Hz per pixel, TA = 4 min 17 s) on different participants to visualize the gains in SNR brought by field strength (Extended Data Fig. 1). Using a longer TE at 3 and 7 T could increase the $T_2^*$ contrast but at the expense of SNR. The signal yet was robust with respect to flip angle variations given the slow increase of $T_1$ versus field strength, making the Ernst angle relatively constant. Acquisitions were performed with 1Tx (body coil)-32Rx, 8Tx-32Rx and 8Tx-32Rx head coils at 3, 7 and 11.7 T, respectively. No bias field correction was performed. Finally, additional $T_2^*$-weighted 2D GRE acquisitions were performed with the same RF coils and RF pulse design strategies to attempt matching the contrast-to-noise ratio at 3, 7 and 11.7 T by adjusting the TE, FA and resolution, while keeping the acquisition time the same (4 min 17 s). The results are presented in Fig. 2. The retained parameters were FA, TR and TE of 27°, 750 ms and 45 ms; 34°, 950 ms and 25 ms; 27°, 600 ms and 20 ms at 3, 7 and 11.7 T, respectively) and spatial resolution (0.5, 0.325 and 0.2-mm in-plane resolution for 3, 7 and 11.7 T, respectively, all with 1-mm slice thickness.

### Biological protocol and results

This study (IDRCB 2022-A02321-42) was approved by the French 'Agence Nationale de Sûreté et du Médicament' regulatory body and a national ethics committee. Written informed consent was obtained from all volunteers. In addition to the usual counter-indications for MRI that constitute exclusion criteria (pacemakers, implants, metallic objects, tattoos, claustrophobia and so on), one notable inclusion criterion was the age of the volunteers, who were between 18 and 40 years to reduce inter-participant variability in genotoxicity test results. Twenty volunteers (24.2 ± 4.8 years old, eight males) were scanned at 11.7 T and the tests mentioned below were performed. Another set of 20 volunteers (23.6 ± 6.0 years old, seven males) were exposed to a 0-T, nocebo, field and underwent the same tests (except genotoxicity). For the latter control group, the environment was the same 11.7-T scanner but with the magnetic field ramped down. All volunteers received the same instructions. To minimize the number of field ramps for the magnet, the study first included the 0-T group, followed by the 11.7-T group. Volunteers received a financial compensation for their participation.

Anxiety of the participants was first evaluated with a questionnaire, whereby a score above 20 on the Hamilton scale constituted an exclusion criterion. The participants underwent cognitive (before, during and after the MRI exam) and vestibular (before and after) tests. The cognitive tests inside the MRI scanner were performed with and without gradient noise to aim at isolating a potential impact of the loud acoustic noise on the results. Blood samples were drawn before and after the exams of the 11.7-T group to conduct a genotoxicity analysis, subcontracted to an external and certified laboratory (GenEvolutioN, Porcheville, France), and look for a potential effect due to exposure of the strong magnetic field[16]. Arterial pressure and cardiac pulsation were measured before, during and after the MRI exam.

**Genotoxic tests.** Genotoxicity testing is based on the detection of chromosome damage in human cells. An alternative to measuring structural aberrations in mitotic cells is to measure micronuclei. These are produced from whole chromosomes or acentric fragments that are unable to attach to the spindle at mitosis and appear during the next interphase as small darkly staining bodies adjacent to the main daughter nucleus. Cytochalasin B (Cyto-B), if added to cultures, inhibits cytokinesis (cell division) but not karyokinesis (nuclear division) resulting in the formation of binucleate cells[41] including micronuclei. Having shown that the repetitive exposure to 3 T in MRI[16] induces mainly terminal deletions, the micronuclei protocol was performed with cytochalasin B addition at 68 h after lymphocyte growth stimulation and with gamma irradiation (IRCM irradiation platform, CEA, Fontenay-aux-Roses, France) as

positive controls. The genotoxicity test was performed by scoring the proportions of micronucleated binucleate cells for each volunteer exposed at 11.7 T compared to the proportion in unexposed controls corresponding to the same volunteer by using Fisher's exact test[42]. Probability values of $P \leq 0.05$ were accepted as significant.

**Behavioral tests.** To assess participants' executive functions, we used the attentional network task[20,21], a model designed to evaluate attentional focus capacity. In a sequence of trials, participants were instructed to promptly indicate the direction (left or right) of a target arrow presented on a computer screen. The target arrow was flanked by additional arrows that could either all align in the same direction as the target (congruent condition) or point toward the opposite direction (incongruent condition). Furthermore, in certain trials, advance cues provided information about the timing and/or location of the impending target.

Participants underwent this task on four occasions: before entering the scanner (Run '1_out'), within the scanner with no noise (Run '2_in'), inside the scanner with the added noise of a magnetic resonance sequence (Run '3_in_noise') and outside the scanner (Run '4_out'). Evaluations outside the scanner were conducted in a soundproofed room, with participants seated before a computer screen displaying the stimuli. Within the scanner, stimuli were visible through a mirror mounted on the head coil, with projections onto an LCD screen positioned at the rear of the scanner.

**Balance tests.** The participants' static balance was assessed before and approximately 25 min after the magnetic field exposure (0 or 11.7 T). A sensation questionnaire was filled after exiting the scanner in another room and before the vestibular test. The volunteers stood for 30 s, first with their eyes open, then with eyes closed, on a force platform equipped with dedicated sensors and connected to a computer and software (AbilyCare, Paris, France) returning a stability score (ranging from 0 to 99)[19].

**Results.** In all tests performed throughout the protocol, no significant differences between the two groups exposed at 11.7 and 0 T could be identified (Extended Data Fig. 2 and Extended Data Table 1). No statistically significant differences in micronuclei count for genotoxicity for each participant before versus after 1 h 30 min exposure at 11.7 T were found either. Follow-up phone calls by the NeuroSpin medical team up to 1 week after exposure at 11.7 T did not indicate any abnormality. Six and four out of 20 volunteers exposed to the 0 and 11.7 T field, respectively, reported fatigue the same day or the day after. Four and one volunteers respectively belonging to the 0 and 11.7-T group reported headaches. Notably different sensations experienced between the two groups involved transient dizziness when entering and exiting the magnet, as well as a metallic taste in the mouth, which are known minor side effects occurring when moving in a magnetic field.

### Reporting summary
Further information on research design is available in the Nature Portfolio Reporting Summary linked to this article.

## Data availability
All images supporting the findings of this study are available via figshare at https://doi.org/10.6084/m9.figshare.25867735.v1 (ref. 43). All behavioral data supporting the findings of this study are available at https://github.com/chrplr/prems. Source data are provided with this paper.

## Code availability
The Attentional Network Task experiment was run with a custom Python code available at https://github.com/chrplr/Posner_attention_networks_task. Behavioral data were analyzed with custom code in the R (v.4.4.1) language available at http://github.com/chrplr/prems. The Python codes are distributed under the Creative Commons 4.0 ShareAlike license (https://creativecommons.org/licenses/by-sa/4.0/legalcode.txt).

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

## Acknowledgements
This project received funding from Banque Publique d'Investissement (BPIFrance) (N.B., A.A., M.L., C.P., L.S., M.B., A.V., D.L.B.), AROMA H2020 FET-Open (grant no. 885876) (N.B., A.A., V.G., C.L.S., A.V., F.M.) and Agence Nationale de la Recherche (ANR) (France 2030 Future Investment Program, grant no. ANR-21-ESRE-0006, ESR/EquipEx+, PRESENCE project) (N.B., F.M., V.G., A.A., C.L.S., M.L., C.P., L.S., M.B., A.V., D.L.B.). We thank the NeuroSpin platform personnel for their support, especially C. Ginisty and L. Laurier, without whom the experiments would not have been possible. We are also immensely grateful to C. Rabrait, A. Mazouyès and F. Lethimonnier who were Iseult's project managers for NeuroSpin, and to the Irfu department at CEA for designing and commissioning the 11.7T magnet, especially P. Védrine, T. Schild and L. Quettier who were project managers for the

magnet and its installation. Siemens Healthineers and GE-Alstom are also thanked for their collaboration.

## Author contributions

N.B., F.M., V.G., A.A., C.L.S., A.M., C.P., L.S., M.B., A.V. and D.L.B. conceived and designed the study. N.B., F.M., V.G., A.A., C.L.S., A.M., M.L., C.P., L.S., M.B. and A.V. conducted the experiments. N.B., F.M., V.G., A.A., C.L.S., A.M., M.L., C.P., L.S., M.B., A.V. and D.L.B. performed the analysis and interpretation of the data. N.B. and D.L.B. wrote the manuscript with input from all authors. D.L.B. conceived and steered the Iseult project. N.B. is the current Iseult project manager.

## Competing interests

N.B. (US20190252788A1, WO2022194711A1), F.M. (WO2022194711A1), V.G. (US20190252788A1, WO2022194711A1) and A.A. (US9291691B2) hold several patents related to this work (pTx). A.M. is an employee of Siemens Healthineers. The other authors declare no competing interests.

## Additional information

**Extended data** is available for this paper at https://doi.org/10.1038/s41592-024-02472-7.

**Correspondence and requests for materials** should be addressed to Denis Le Bihan.

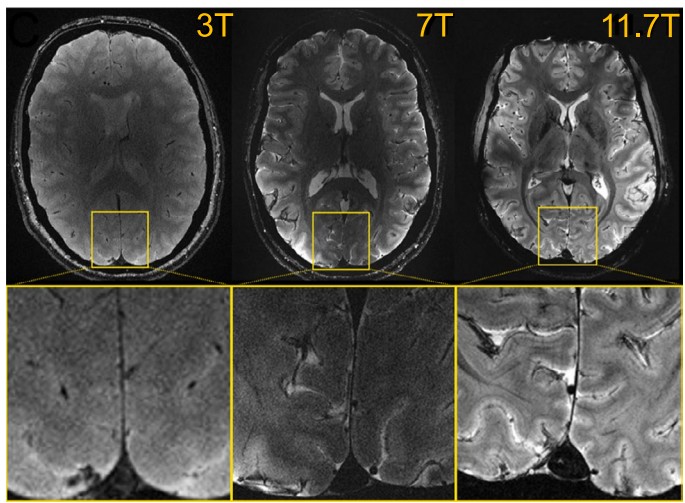

**Extended Data Fig. 1 | T$_2^*$-weighted 2D GRE axial images acquired at 3T, 7T and 11.7T (different subjects).** Acquisitions were performed with resolution = 0.2 × 0.2 × 1 mm$^3$, FA = 27°, TE = 20 ms, TR = 0.6 s, bandwidth = 40 Hz/pixel, acquisition time = 4 min 20 s.

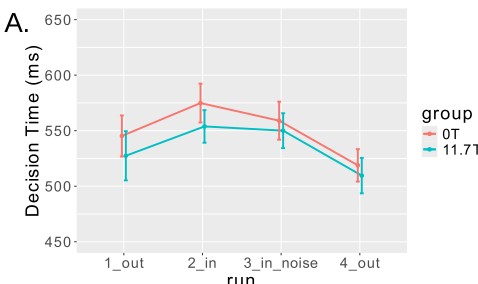

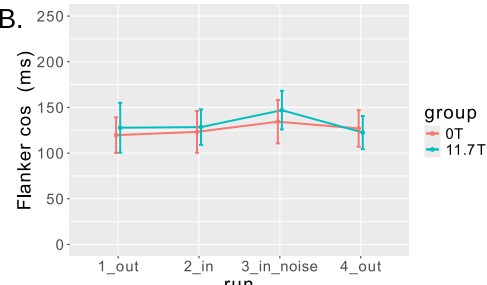

**Extended Data Fig. 2 | Results of the Attentional Network Task.** The number of samples is n = 20 for both 0T and 11.7T groups. Data are presented as mean values ± 1 standard error. Panel **A** (left). Mean decision times as a function of Run and Group. An Analysis of Variance with the factors Group and Run show no significant effect of Group (p = 0.54) and not any interaction (p = 0.62)

(Extended Data Table 2). Both groups were slower inside the scanner than outside (Effect of Run: p < 0.001). Panel **B** (right). Flanker cost (difference in decision-times between incongruent and congruent trials). In an Analysis of Variance, the effect of Group was non significant (p = 0.57), nor interacted with any other factor (all p-values > 0.5) (Extended Data Table 3).

**Extended Data Table 1 | Results of the balance tests**

| Group | Test results | | | |
|---|---|---|---|---|
| | **Eyes closed** | | **Eyes opened** | |
| | **Before** | **After** | **Before** | **After** |
| 0T | 98.7 ± 1.0 | 98.3 ± 1.3 | 98.8 ± 0.9 | 98.5 ± 1.8 |
| 11.7T | 99.0 ± 0.0 | 98.7 ±1.0 | 98.7 ± 1.2 | 98.9 ± 0.3 |

| | Anova | | | |
|---|---|---|---|---|
| **Effect** | **df** | **F** | **ges** | **p.value** |
| group | 1, 38 | 1.66 | .014 | .206 |
| test | 2, 38 | 1.26 | .016 | .422 |
| group:test | 2, 38 | 1.26 | .010 | .610 |

Measurements were performed before entering the scanner and 25 minutes after exiting, with eyes closed, then with eyes opened. The scores represent the probability of belonging to a group of participants without balance issues[19]. The top part of the table reports the descriptive statistics (means and standard errors with n = 20 for both 0T and 11.7T groups), the bottom part reports the result of an analysis of variance (df=degrees of freedom; ges=Generalized Eta Squared).

**Extended Data Table 2 | Results of the Attentional Network Task: Mean decision times**

| Effect | df | MSE | F | ges(*) | p.value |
|---|---|---|---|---|---|
| group | 1, 37 | 20917.77 | 0.38 | .009 | .541 |
| run | 2.26, 83.67 | 925.50 | 27.21 *** | .063 | <.001 |
| group:run | 2.26, 83.67 | 925.50 | 0.52 | .001 | .620 |
| (*) ges=Generalized Eta-Squared measure of effect size | | | | | |

Results from an Analysis of Variance (performed with the aov_var function of the R software) with the factors Group and Run on the individual decision times in the Attentional Network Task (data displayed on panel A of Fig. 2).

**Extended Data Table 3 | Results of the Attentional Network Task: Flanker cost**

| Effect | df | MSE | F | ges(*) | p.value |
|---|---|---|---|---|---|
| group | 1, 37 | 43570.87 | 0.33 | .007 | .567 |
| flanker_congruency | 1, 37 | 4854.37 | 266.18 *** | .392 | <.001 |
| group:flanker_congruency | 1, 37 | 4854.37 | 0.12 | <.001 | .736 |
| run | 2.21, 81.64 | 2006.16 | 27.89 *** | .058 | <.001 |
| group:run | 2.21, 81.64 | 2006.16 | 0.40 | <.001 | .690 |
| flanker_congruency:run | 2.55, 94.21 | 488.73 | 3.01 * | .002 | .042 |
| group:flanker_congruency:run | 2.55, 94.21 | 488.73 | 0.61 | <.001 | .581 |

(*) ges=Generalized Eta-Squared measure of effect size

Results from an Analysis of Variance (performed with the aov_car function of R software) with the factors Group, Run and Flanker_Congruency (congruent vs. Incongruent) on the decision times in the Attentional Network Task (data displayed on panel B of Fig. 2).

# Reporting Summary

## Statistics

For all statistical analyses, confirm that the following items are present in the figure legend, table legend, main text, or Methods section.

| n/a | Confirmed | |
|---|---|---|
| ☐ | ☒ | The exact sample size (*n*) for each experimental group/condition, given as a discrete number and unit of measurement |
| ☐ | ☒ | A statement on whether measurements were taken from distinct samples or whether the same sample was measured repeatedly |
| ☐ | ☒ | The statistical test(s) used AND whether they are one- or two-sided<br>*Only common tests should be described solely by name; describe more complex techniques in the Methods section.* |
| ☒ | ☐ | A description of all covariates tested |
| ☒ | ☐ | A description of any assumptions or corrections, such as tests of normality and adjustment for multiple comparisons |
| ☐ | ☒ | A full description of the statistical parameters including central tendency (e.g. means) or other basic estimates (e.g. regression coefficient) AND variation (e.g. standard deviation) or associated estimates of uncertainty (e.g. confidence intervals) |
| ☐ | ☒ | For null hypothesis testing, the test statistic (e.g. *F*, *t*, *r*) with confidence intervals, effect sizes, degrees of freedom and *P* value noted<br>*Give P values as exact values whenever suitable.* |
| ☒ | ☐ | For Bayesian analysis, information on the choice of priors and Markov chain Monte Carlo settings |
| ☒ | ☐ | For hierarchical and complex designs, identification of the appropriate level for tests and full reporting of outcomes |
| ☐ | ☒ | Estimates of effect sizes (e.g. Cohen's *d*, Pearson's *r*), indicating how they were calculated |

*Our web collection on statistics for biologists contains articles on many of the points above.*

## Software and code

Policy information about availability of computer code

| Data collection | the Attentional Network Task experiment was ran with a custom python code available at https://github.com/chrplr/ Posner_attention_networks_task (note: "GRAPE pulse" in not a code, but the name of a standard MRI sequence) |
|---|---|
| Data analysis | behavioral data were analysed with custom code in the R language available at http://github.com/chrplr/prems |

For manuscripts utilizing custom algorithms or software that are central to the research but not yet described in published literature, software must be made available to editors and reviewers. We strongly encourage code deposition in a community repository (e.g. GitHub). See the Nature Portfolio guidelines for submitting code & software for further information.

## Data

Policy information about availability of data

All manuscripts must include a data availability statement. This statement should provide the following information, where applicable:

- Accession codes, unique identifiers, or web links for publicly available datasets
- A description of any restrictions on data availability
- For clinical datasets or third party data, please ensure that the statement adheres to our policy

All behavioral data supporting the findings of this study are available at http://github.com/chrplr/prems

## Human research participants

Policy information about studies involving human research participants and Sex and Gender in Research.

| | |
|---|---|
| Reporting on sex and gender | No sex or gender analysis was performed. |
| Population characteristics | Group 0T : n=20 24.2 ± 4.8 year 8 males<br>Group 11.7T : n=20 23.6 ± 6.0 year 7 males |
| Recruitment | Subjects were recruited freely through local annoucements. No bias has to be reported. |
| Ethics oversight | This study was approved by the French "Agence Nationale de Sûreté et du Médicament" (ANSM) regulatory body and a national ethics committe (IDRCB 2022-A02321-42) as stated in the manuscript. |

Note that full information on the approval of the study protocol must also be provided in the manuscript.

# Field-specific reporting

Please select the one below that is the best fit for your research. If you are not sure, read the appropriate sections before making your selection.

☒ Life sciences ☐ Behavioural & social sciences ☐ Ecological, evolutionary & environmental sciences

For a reference copy of the document with all sections, see nature.com/documents/nr-reporting-summary-flat.pdf

# Life sciences study design

All studies must disclose on these points even when the disclosure is negative.

| | |
|---|---|
| Sample size | The sample size was determined by the regulation agency approval to test MRI at 11.7T (world premiere). 20 subjects were exposed to 11.7T and 20 subjects to 0T. Subjects were not aware of the presence or not of the magnetic field. This is stated in the manuscript. |
| Data exclusions | All included subjects are analyzed. Exclusion criteria (as stated in the Methods section of the manuscript) were: usual counter-indications for MRI (pacemakers, implants, metallic objects, tattoos, claustrophobia etc), age (below 18 and above 40yo), anxiety (Hamilton) score >20 |
| Replication | N/A. This study allowed only 20 subjects to be scanned, as mentioned in the manuscript. |
| Randomization | N/A Data acquisitions were performed in the order subjects came to be enrolled as volunteers for this study. The first 20 subjects were "scanned" at 0T, when the magnet was off (during several weeks). The next 20 subjects were scanned at 11.7T when the magnet was on. Subjects were not aware of the field value (0 or 11.7T), as stated in the manuscript. |
| Blinding | N/A. For the above reason blinding was not possible for the data acquisition. |

# Reporting for specific materials, systems and methods

We require information from authors about some types of materials, experimental systems and methods used in many studies. Here, indicate whether each material, system or method listed is relevant to your study. If you are not sure if a list item applies to your research, read the appropriate section before selecting a response.

### Materials & experimental systems

| n/a | Involved in the study |
|---|---|
| ☒ | ☐ Antibodies |
| ☒ | ☐ Eukaryotic cell lines |
| ☒ | ☐ Palaeontology and archaeology |
| ☒ | ☐ Animals and other organisms |
| ☒ | ☐ Clinical data |
| ☒ | ☐ Dual use research of concern |

### Methods

| n/a | Involved in the study |
|---|---|
| ☒ | ☐ ChIP-seq |
| ☒ | ☐ Flow cytometry |
| ☐ | ☒ MRI-based neuroimaging |

# Magnetic resonance imaging

## Experimental design

| | |
|---|---|
| Design type | Structural MRI |
| Design specifications | N/A (not a fMRI study). |
| Behavioral performance measures | N/A (only structural MRI was performed, as stated in the manuscript). |

## Acquisition

| | |
|---|---|
| Imaging type(s) | Structural |
| Field strength | 11.7T |
| Sequence & imaging parameters | All details are provided in the Methods section |
| Area of acquisition | Head |

Diffusion MRI ☐ Used ☒ Not used

## Preprocessing

| | |
|---|---|
| Preprocessing software | N/A. It is a structural MRI study |
| Normalization | *If data were normalized/standardized, describe the approach(es): specify linear or non-linear and define image types used for transformation OR indicate that data were not normalized and explain rationale for lack of normalization.* |
| Normalization template | *Describe the template used for normalization/transformation, specifying subject space or group standardized space (e.g. original Talairach, MNI305, ICBM152) OR indicate that the data were not normalized.* |
| Noise and artifact removal | *Describe your procedure(s) for artifact and structured noise removal, specifying motion parameters, tissue signals and physiological signals (heart rate, respiration).* |
| Volume censoring | *Define your software and/or method and criteria for volume censoring, and state the extent of such censoring.* |

## Statistical modeling & inference

| | |
|---|---|
| Model type and settings | N/A. It is a structural MRI study |
| Effect(s) tested | *Define precise effect in terms of the task or stimulus conditions instead of psychological concepts and indicate whether ANOVA or factorial designs were used.* |

Specify type of analysis: ☐ Whole brain ☐ ROI-based ☐ Both

| | |
|---|---|
| Statistic type for inference (See Eklund et al. 2016) | *Specify voxel-wise or cluster-wise and report all relevant parameters for cluster-wise methods.* |
| Correction | *Describe the type of correction and how it is obtained for multiple comparisons (e.g. FWE, FDR, permutation or Monte Carlo).* |

## Models & analysis

| n/a | Involved in the study |
|---|---|
| ☒ ☐ | Functional and/or effective connectivity |
| ☒ ☐ | Graph analysis |
| ☒ ☐ | Multivariate modeling or predictive analysis |

