## [Peer Review File · Nature Methods]

First in vivo images of the human brain revealed with the Iseult 11.7T MRI scanner

Corresponding Author: Professor Denis Le Bihan

Version 0:

Decision Letter:

18th Jan 2024

Dear Professor Le Bihan,

Thank you for your inquiry about submitting your manuscript, "First in vivo images of the human brain revealed with the Iseult 11.7T MRI scanner" to Nature Methods. The paper sounds like it may be of interest, and should fit the scope of the journal. We would be willing to consider it for publication in Nature Methods.

Of course, it is very difficult to judge a paper based only on the limited information available in a presubmission inquiry. Therefore I am sure you understand that we cannot promise to send your paper out for peer review and must read it in its entirety before deciding if this would be suitable.

Please keep in mind that the journal is aimed at a large, interdisciplinary audience and places a strong emphasis on the practical value of the work presented for basic research in the life sciences. We strongly encourage you to include data to validate method performance and demonstrate its general applicability.

You will find our Guide to Authors at <http://www.nature.com/naturemethods> to assist you in preparing your manuscript. However, it is not necessary at this stage to spend major effort adhering to our detailed formatting instructions.

Thank you for your interest in Nature Methods.

Best regards,
Nina

Nina Vogt, PhD
Senior Editor
Nature Methods

Version 1:

Decision Letter:

13th Feb 2024

Dear Professor Le Bihan,

Thank you for submitting your manuscript entitled "First in vivo images of the human brain revealed with the Iseult 11.7T MRI scanner". We have given the paper our careful consideration and find it of potential interest. However, we are concerned that

sending the current manuscript out to review could lead to unnecessary delays and possibly an undesirable outcome of the review process.

In particular, the manuscript currently reads like a press release rather than a research paper. Instead of focusing on the history of the project, please do describe the technology and its capabilities (without unnecessary hype). For examples of similar papers, please see <https://www.nature.com/articles/s41592-023-02068-7> and <https://www.nature.com/articles/s41592-021-01317-x>.

We therefore invite you to revise your manuscript to address these concerns before we make a final determination on whether to send your manuscript for external peer-review. Please ensure that the revised version is as concise as possible, and that it conforms to our format requirements (see <http://www.nature.com/nmeth> for our Guide to Authors).

We shall hope to receive your revised version as soon as you are able to complete the suggested revisions. If something similar is published in the interim we will have to consider the impact it has on the novelty of the revised manuscript.

If you anticipate a delay of more than four weeks, please let us know. In this event, we will still be happy to reconsider your paper at a later date so long as nothing similar has been accepted for publication at Nature Methods or published elsewhere. In the event of publication, however, the received date would be that of the revised rather than the original version.

If you are not interested in submitting a revised manuscript in the future please let me know immediately so we can close your file. If you have any questions, please contact me.

Please use the link below when you are prepared to resubmit.

Link Redacted

Note: The above URL links to your confidential home page and associated information about manuscripts you may have submitted, or that you are reviewing for us. If you wish to forward this email to co-authors, please delete the link to your homepage.

Thank you for your interest in Nature Methods.

Best regards,
Nina

Nina Vogt, PhD
Senior Editor
Nature Methods

** For Nature Portfolio general information and news for authors, see <http://npg.nature.com/authors>.

Version 2:

Decision Letter:

24th Apr 2024

Dear Professor Le Bihan,

Thank you for your patience. Your Brief Communication, "First in vivo images of the human brain revealed with the Iseult 11.7T MRI scanner", has now been seen by three reviewers. As you will see from their comments below, although the reviewers find your work of considerable potential interest, they have raised a number of concerns. We are interested in the possibility of publishing your paper in Nature Methods, but would like to consider your response to these concerns before we reach a final decision on publication.

We therefore invite you to revise your manuscript to address these concerns. Specifically, we recommend that you show different contrasts. Furthermore, the data should be made available publicly. For your information, I'd like to draw your attention to our Figshare initiative (<https://www.springernature.com/gp/authors/research-data/figshare-integration>).

Link Redacted

We hope to receive your revised paper within 2-3 months. If you cannot send it within this time, please let us know. In this event, we will still be happy to reconsider your paper at a later date so long as nothing similar has been accepted for publication at Nature Methods or published elsewhere.

OPEN SCIENCE REQUIREMENTS

REPORTING SUMMARY AND EDITORIAL POLICY CHECKLISTS

DATA AVAILABILITY

All novel DNA and RNA sequencing data, protein sequences, genetic polymorphisms, linked genotype and phenotype data, gene expression data, macromolecular structures, and proteomics data must be deposited in a publicly accessible database, and accession codes and associated hyperlinks must be provided in the "Data Availability" section.

To further increase transparency, we encourage you to provide, in tabular form, the data underlying the graphical representations used in your figures. This is in addition to our data-deposition policy for specific types of experiments and large

datasets. For readers, the source data will be made accessible directly from the figure legend. Spreadsheets can be submitted in .xls, .xlsx or .csv formats. Only one (1) file per figure is permitted: thus if there is a multi-paneled figure the source data for each panel should be clearly labeled in the csv/Excel file; alternately the data for a figure can be included in multiple, clearly labeled sheets in an Excel file. File sizes of up to 30 MB are permitted. When submitting source data files with your manuscript please select the Source Data file type and use the Title field in the File Description tab to indicate which figure the source data pertains to.

CODE AVAILABILITY

Please include a "Code Availability" subsection in the Online Methods which details how your custom code is made available. Only in rare cases (where code is not central to the main conclusions of the paper) is the statement "available upon request" allowed (and reasons should be specified).

MATERIALS AVAILABILITY

ORCID

Nature Methods is committed to improving transparency in authorship. As part of our efforts in this direction, we are now requesting that all authors identified as 'corresponding author' on published papers create and link their Open Researcher and Contributor Identifier (ORCID) with their account on the Manuscript Tracking System (MTS), prior to acceptance. This applies to primary research papers only. ORCID helps the scientific community achieve unambiguous attribution of all scholarly contributions. You can create and link your ORCID from the home page of the MTS by clicking on 'Modify my Springer Nature account'. For more information please visit <http://www.springernature.com/orcid>.

Best regards,
Nina

Nina Vogt, PhD
Senior Editor
Nature Methods

Reviewers' Comments:

Reviewer #1:

Remarks to the Author:

For the last four decades, humanity has made significant strides in uncovering the complexities of the human brain. This progress has been made possible through the development of advanced hardware and systematic methods applied to increasingly sophisticated questions. In their manuscript titled "First in vivo images of the human brain revealed with the Iseult

11.7T MRI scanner," Boulant et al. detail a remarkable technological achievement that brought techniques previously used in small animals (<https://www.kennedykrieger.org/kirby-research-center/preclinical-facility/facilities-and-instruments/bruker-biospec-11-7-t>) to the human brain. The contribution is unparalleled, and the results are unprecedented in terms of resolution and signal-to-noise ratio.

In order to maximize the impact of this scientific report, it is important to provide open access to the initial data presented in figure 2. This will pique the curiosity of the community and greatly increase the number of citations for this study. Although the authors have stated that "All data supporting the findings of this study are available within the paper and its Extended Method," I was unable to locate this information in the materials provided for review. Sharing data is now more accessible than ever through platforms like neurovault.org, which I highly recommend to the authors.

In essence, sharing the neuroimaging data displayed in figure 2 will enable the community to adopt this technological advancement with enthusiasm, scrutinize the accuracy of the findings, and pave the way for future research and discoveries. I strongly urge you to consider sharing this valuable data.

Reviewer #2:

Remarks to the Author:

This manuscript describes the first MR images of the human brain in vivo at a magnetic field strength of 11.7 Tesla. This is the culmination of more than 20 years of development across a wide range of aspects (MR hardware including main magnet design and RF coil engineering; pulse design for parallel radiofrequency (RF) transmit (pTx); MR sequence development; tests of physiological interaction). It is a unique achievement and I want to congratulate the authors and the consortium that they have persevered and achieved this key milestone!

In this manuscript the authors show high resolution MR images for different MR contrasts of a single subject (T2- and T2*-weighted) and compare them to images at 3T and 7T. The images are impressive, in particular the relatively low inhomogeneity of the T2-weighted images using the custom-made RF coil and RF pulse design, which shows the potential of 11.7T MR imaging. They have also performed a range of physiological and cognitive tests in a group of 20 healthy volunteers which show no significant effect of field strength between no field and 11.7T indicating that humans can tolerate this very high magnetic field strength.

There are a few points that would need to be addressed before I would recommend publication, particularly considering the claimed potential of going higher and higher with regard to field strength. While I acknowledge that this manuscript demonstrates the first results, I find it disappointing that (in terms of imaging results) only images of two MR contrast of a single subject are presented here. Recent reviews and opinion pieces have mentioned a range of methods and applications that would profit from ultra-high field strengths beyond 7T (and 9.4T), see e.g. recent Special Issue of Magnetic Resonance Materials in Physics, Biology and Medicine, Vol36(2). It would be important to demonstrate more rigorously how this development is getting closer to the expectations. This should include:

- 1) While the image resolution is certainly impressive, similar resolutions have been demonstrated using lower field strengths (7T), albeit with longer scan times, see for example Feinberg et al, "Next generation MRI scanner" Nature Methods 20, 2023. doi: 10.1038/s41592-023-02068-7.
- 2) Show images of all subjects, e.g. in the supplementary materials, to demonstrate the robustness of results or at least discuss issues with robustness at this stage.
- 3) Show results of other relevant contrasts (T1, DWI, MR angiography) and methods that are very likely to profit from 11.7T (e.g. MRS, fMRI) as well, or at least discuss issues that arise with these and other important MR contrasts.
- 4) While I understand that the authors have 3T and 7T scanners available at their facility, I would argue that the relevant reference point would be the human imaging results obtained at the field strength of 10.5T. A more detailed comparison with published findings at this field strength (closest to the one used here) would be highly relevant.
- 5) The comparison with 3T and 7T results seems biased as it is unclear if the parameters used were optimized at the lower field strengths. Acquisition parameters have not been provided, only stated that "similar scans were performed".
- 6) While the authors briefly mention next steps in the technical developments, I miss a more detailed discussion of how much these will improve SNR (compared to theoretical possible SNR) and resolution (e.g. with the improved motion mitigation strategies) and other aspects.
- 7) I miss a clearer and more detailed discussion about what the current results and future developments mean for scientific/neuroscience and potentially clinical applications.

Minor points:

- 8) A few times ISMRM abstracts are used as references, but these seem only accessible to registrants of the conference. While most people in the field will likely have/get access, this is not the case for the average reader.
- 9) The project name Iseult is used throughout the manuscript, but it is unclear what the specific meaning is?

Reviewer #3:

Remarks to the Author:

This groundbreaking work by Boulant et al. marks a significant milestone in ultrahigh field MRI – the first in-vivo human brain images at 11.7T. Prior to this, the pinnacle of human MRI field strength stood at 10.5T (but to my knowledge body-only), and 9.4T for brain; reaching 11.7T in the brain represents a remarkable leap forward in technological advancement. The images obtained from this Iseult magnet exhibit remarkable quality and extremely high resolution was obtained at reasonable scanning time. In addition, the feasibility of keeping a living and breathing human being under 11.7T field without significant physiological symptoms is shown and is an important and encouraging step towards more widespread application of ultrahigh field MRI, with its great potential for resolving hitherto unexplored features of the human brain.

The originality and novelty of the work is as high as can be, and indeed the work is well written and worthy of being presented in the most prestigious of venues.

I only have a few comments that require some attention:

1. I think the work would benefit from a better description of state-of-the-art resolution compared to the new scanner.
2. In that sense I was also somewhat surprised that the authors didn't attempt to break any "records" for resolution. For instance, doi: 10.1016/j.neuroimage.2013.08.013 acquired images at 9.4T with 0.13x0.13x0.8 mm³ and doi: 10.1371/journal.pone.0133921 presented 0.44 mm isotropic resolution. Without robbing any novelty from the current work, it may be beneficial to at least discuss these previous works and state the highest state-of-the-art resolutions in other fields (perhaps also vis-a-vis scan time and motion effects).
3. In Fig 2 it is unclear which exact conditions were used in each of the 3/7/11.7 T measurements. Would they be expected to only vary in resolution (i.e. were TE/TR adjusted for the different field T2^{*}/T1) or were they performed under the same conditions (TE/TR)? If the latter, contrast is expected to vary, and to make a good comparison it is recommended to match the contrast conditions while only varying resolution if possible. Please clarify.
4. Line 272 TE = 301ms is correct??

Author Rebuttal letter:

Rebuttal letter 1:

RESPONSES TO THE REVIEWERS COMMENTS

Reviewer #1:

Remarks to the Author:

For the last four decades, humanity has made significant strides in uncovering the complexities of the human brain. This progress has been made possible through the development of advanced hardware and systematic methods applied to increasingly sophisticated questions. In their manuscript titled "First in vivo images of the human brain revealed with the Iselt 11.7T MRI scanner," Boulant et al. detail a remarkable technological achievement that brought techniques previously used in small animals <https://www.kennedykrieger.org/kirby-research-center/preclinical-facility/facilities-and-instruments/bruker-biospec-11-7-t>) to the human brain. The contribution is unparalleled, and the results are unprecedented in terms of resolution and signal-to-noise ratio.

Thanks a lot for your great comments! We, indeed, also house a 17.2T MRI scanner from Bruker which we have used for microscopic, single neuron imaging, in the Aplysia (<https://www.nature.com/articles/s41598-017-05586-5>, <https://doi.org/10.1073/pnas.1403739111>)

In order to maximize the impact of this scientific report, it is important to provide open access to the initial data presented in figure 2. This will pique the curiosity of the community and greatly increase the number of citations for this study. Although the authors have stated that "All data supporting the findings of this study are available within the paper and its Extended Method," I was unable to locate this information in the materials provided for review. Sharing data is now more accessible than ever through platforms like neurovault.org, which I highly recommend to the authors. In essence, sharing the neuroimaging data displayed in figure 2 will enable the community to adopt this technological advancement with enthusiasm, scrutinize the accuracy of the findings, and pave the way for future research and discoveries. I strongly urge you to consider sharing this valuable data.

We fully agree. As suggested, we have made our images data available on the Figshare platform.

Reviewer #2:

Remarks to the Author:

This manuscript describes the first MR images of the human brain in vivo at a magnetic field strength of 11.7 Tesla. This is the culmination of more than 20 years of development across a wide range of aspects (MR hardware including main magnet design and RF coil engineering; pulse design for parallel radiofrequency (RF) transmit (pTx); MR sequence development; tests of physiological interaction). It is a unique achievement and I want to congratulate the authors and the consortium that they have persevered and achieved this key milestone!

Thanks a lot for your great comments

In this manuscript the authors show high resolution MR images for different MR contrasts of a single subject (T2- and T2*-weighted) and compare them to images at 3T and 7T. The images are impressive, in particular the relatively low inhomogeneity of the T2-weighted images using the custom-made RF coil and RF pulse design, which shows the potential of 11.7T MR imaging. They have also performed a range of physiological and cognitive tests in a group of 20 healthy volunteers which show no significant effect of field strength between no field and 11.7T indicating that humans can tolerate this very high magnetic field strength.

There are a few points that would need to be addressed before I would recommend publication, particularly considering the claimed potential of going higher and higher with regard to field strength. While I acknowledge that this manuscript demonstrates the first results, I find it disappointing that (in terms of imaging results) only images of two MR contrast of a single subject are presented here. Recent reviews and opinion pieces have mentioned a range of methods and applications that would profit from ultra-high field strengths beyond 7T (and 9.4T), see e.g. recent Special Issue of Magnetic Resonance Materials in Physics, Biology and Medicine, Vol36(2). It would be important to demonstrate more rigorously how this development is getting closer to the expectations.

We can only agree with your comments. Please note that the results of Figure 2 are not from the same subject, as was stated in the figure caption. The reason why we could not pursue more acquisitions is that we had been authorized by the French regulatory agency to scan only a small cohort (20 subjects) to validate the overall 11.7T MRI set-up and safety. Obviously, we want to do much more, but this will take months and we wanted to have a fast account on this first step, hence the Brief Communication format. This study was exploratory and required iterations to converge towards the results we present. We have clarified this point in the manuscript. Motion was also an issue. Diffusion and fMRI are also future investigations but more tool developments are necessary. The T2 and T2* weighted acquisitions explored two contrasts based on gradient echo and spin echo mechanisms, thus two different RF pulse categories. It was impossible to cover all contrasts in this first 20 subjects study. Future work will address exploration of more contrasts. As suggested, we have outlined a few potential (clinical and neuroscience) applications of UHF MRI at the end of the manuscript, skipping details, tough, to remain within the limit of the world count.

This should include:

1) While the image resolution is certainly impressive, similar resolutions have been demonstrated using lower field strengths (7T), albeit with longer scan times, see for example Feinberg et al, "Next generation MRI scanner" *Nature Methods* 20, 2023. doi: 10.1038/s41592-023-02068-7.

Thank you. Indeed, high resolution scans are possible of course, at lower field, but require longer acquisition times. This is a crucial point which we aimed to address in Fig.2, showing that by using the same spatial resolution and acquisition time anatomical details are lost at lower fields due to poorer signal and contrast to noise ratio, as was stated in the figure caption. While it is impossible to make fair comparisons unless the same contrasts, sequences, acquisition times etc are the same (their quality naturally depends on many parameters) we have replaced those images with a set of contrast to noise ratio "optimized" images acquired at 3T and 7T showing that similar contrast and acquisition times requires lowering the spatial resolution at lower field strength (the former images have been shifted to Supplementary Materials to remain within the allowed number of figures). Please also note that the high resolution anatomical scans we could find in the mentioned article (which was cited in our manuscript) were: QSM (TA = 7 min, res = $0.21 \times 0.21 \times 1.5$ mm³), T1 and T2 maps with fingerprinting (TA = 4 min but not including the B1 acquisition time needed, res = $(0.56$ mm)³ and time of flight MRA (TA=10 min 19 s, resolution = $(0.4$ mm)³). However, it is true indeed that the magnetic field is not the only possibility to increase performance (SNR), e.g. with large receive coil arrays as discussed in the article mentioned above. This point is mentioned in our manuscript.

2) Show images of all subjects, e.g. in the supplementary materials, to demonstrate the robustness of results or at least discuss issues with robustness at this stage.

The data in Figure 2 comes from several subjects, as was stated in the figure caption. We cannot, unfortunately, comment a lot on the robustness of the results given that the 20 exams authorized by our regulatory body were never exactly the same. We had to iterate with our pulse design methods and parameter optimization to converge towards the results presented. What we witnessed on the other hand was that nearly 2/3 of the high resolution scans were corrupted by motion, which is therefore an important problem to address in the future. We now provide a few more comments regarding those limitations in the manuscript. Work is still needed to increase robustness and deploy

further our portfolio of sequences equipped with the necessary tools.

3) Show results of other relevant contrasts (T1, DWI, MR angiography) and methods that are very likely to profit from 11.7T (e.g. MRS, fMRI) as well, or at least discuss issues that arise with these and other important MR contrasts.

Clearly, it would be interesting, indeed, to see more contrasts. However, given the limited number of exams we could perform (20) and the technical challenges we faced (protocol optimization, motion, RF field inhomogeneity mitigation which highly depends on the particular acquisitions and the RF pulses therein), we chose on purpose for this first protocol to focus on anatomical T2 and T2* weighted acquisitions to secure good quality data. We tried T1-weighted acquisitions as well but faced difficulties regarding pulse inversions, as mentioned in the manuscript. This is currently attributed to more severe RF field inhomogeneity of course, but also because of the RF coil architecture and increased power losses at higher frequencies. Those limitations have been added in the manuscript.

4) While I understand that the authors have 3T and 7T scanners available at their facility, I would argue that the relevant reference point would be the human imaging results obtained at the field strength of 10.5T. A more detailed comparison with published findings at this field strength (closest to the one used here) would be highly relevant.

We agree that it could be a nice addition. The 10.5T, however, remains also unique and we do not have access to the scanner. 10.5T and 11.7T are still not plug and play scanners and we could not reasonably request such a scan without the team present on site to deploy all our parallel transmission tools (we do not have experience with their RF coil either). We would like to argue that spanning 3T-7T-11.7T appears more relevant to us: 3T is the most intense field ubiquitously found in the clinic, 7T is available across about 100 sites and 11.7T is the most powerful now. Lastly, we cannot make comparisons with 10.5T unless the two scanners are benchmarked with identical protocols, receive coil arrays, methods and sequences. Comparing with published results, thus, we fear would not be relevant. Please note however that an SNR measurement was performed at 3T, 7T, 9.4T, 10.5T and 11.7T on a phantom (reference #7: Le Ster et al.) confirming more SNR at 11.7T.

5) The comparison with 3T and 7T results seems biased as it is unclear if the parameters used were optimized at the lower field strengths. Acquisition parameters have not been provided, only stated that "similar scans were performed".

This is a very important point, indeed. We are sorry the caption of the figure was not clear. Our goal was to show the image quality obtained at 3, 7 and 11.7T using the same set-up (sequence parameters, especially spatial resolution and acquisition time, see also our response to point#1). While "quality" might be ill defined, in our view, it all depends on what we want to optimize. Our initial goal was to show that by using the same resolution, acquisition time and contrast type the gain in SNR at 11.7T made details much more visible. Another approach would be to somewhat "optimize" the contrast to noise ratio at the 3 field strengths, while keeping the contrast type and acquisition time similar, showing that the spatial resolution must necessarily be lower at lower field. We are now providing such a comparison in the new Fig.2, but one has to accept that this comparison is arguably not as fair as the SNR comparison (now in the Supplementary Materials section) given that acquisition parameters (flip angle, TE, etc) need to be adjusted to compensate for the changes in T1, T2 and T2* values with field strength while compromising the SNR. The main, practical message is that at 11.7T we get high contrast and high resolution while keeping acquisition time short, which is not possible at lower field strength. We have provided more explicitly the parameters for the 3 and 7T acquisitions, and we elaborate more regarding this choice of comparisons.

6) While the authors briefly mention next steps in the technical developments, I miss a more detailed discussion of how much these will improve SNR (compared to theoretical possible SNR) and resolution (e.g. with the improved motion mitigation strategies) and other aspects.

We have added a few sentences and references regarding this important point, however, we are limited by the word count under the Brief Communication format. Of particular interest is the number of receive channels which should be upgraded in the future, as demonstrated recently by the 10.5T group in Minneapolis. We have added a reference from that group to illustrate this idea.

7) I miss a clearer and more detailed discussion about what the current results and future developments mean for scientific/neuroscience and potentially clinical applications.

We have added a few sentences and references regarding this important point, within the word count limit.

Minor points:

8) A few times ISMRM abstracts are used as references, but these seem only accessible to registrants of the conference. While most people in the field will likely have/get access, this is not the case for the average reader.

We are very sorry, but some of these studies are, indeed, only available in that format. We leave it to the Editor's decision whether they should be removed or not.

9) The project name Iseult is used throughout the manuscript, but it is unclear what the specific meaning is?

The reason is that it was a French-German project and that the father of the project (last author of the paper) is a musician. Hence, in 2004, he coined the overall project "Iseult", which is, after all, more romantic than the acronym INUMAC (Imaging of Neuro disease Using high field MR and Contrastophores) which has also been used occasionally for the project. This anecdotal point has not been added to the manuscript.

Reviewer #3:

Remarks to the Author:

This groundbreaking work by Boulant et al. marks a significant milestone in ultrahigh field MRI – the first in-vivo human brain images at 11.7T. Prior to this, the pinnacle of human MRI field strength stood at 10.5T (but to my knowledge body-only), and 9.4T for brain; reaching 11.7T in the brain represents a remarkable leap forward in technological advancement. The images obtained from this Iseult magnet exhibit remarkable quality and extremely high resolution was obtained at reasonable scanning time. In addition, the feasibility of keeping a living and breathing human being under 11.7T field without significant physiological symptoms is shown and is an important and encouraging step towards more widespread application of ultrahigh field MRI, with its great potential for resolving hitherto unexplored features of the human brain.

The originality and novelty of the work is as high as can be, and indeed the work is well written and worthy of being presented in the most prestigious of venues.

Thanks a lot for your great comments!

I only have a few comments that require some attention:

1. I think the work would benefit from a better description of state-of-the-art resolution compared to the new scanner.

We have added a few sentences and references regarding this important point. However, we are limited by the word count under the Brief Communication format.

2. In that sense I was also somewhat surprised that the authors didn't attempt to break any "records" for resolution. For instance, doi: 10.1016/j.neuroimage.2013.08.013 acquired images at 9.4T with 0.13x0.13x0.8 mm³ and doi: 10.1371/journal.pone.0133921 presented 0.44 mm isotropic resolution. Without robbing any novelty from the current work, it may be beneficial to at least discuss these previous works and state the highest state-of-the-art resolutions in other fields (perhaps also vis-a-vis scan time and motion effects).

Thank you. We are also aware of the work mentioned above and we now reference it. Yet please note that the 0.13x0.13x0.8 mm³ resolution above was performed in 13 minutes, which still remains quite remarkable. But we cannot dissociate resolution and acquisition time. In this first run, our goal above all was to confirm the absence of adverse effects on the volunteers and perform the first high resolution scans at 11.7T to demonstrate the potential of MRI at 11.7T. With 20 chances (volunteers) or iterations, this is the best we were able to achieve. We also tried 0.175 × 0.175 × 1 mm³ but as mentioned above, the results turned corrupted by motion. Obviously, we want to do much more, but this will take months, more work and subjects, and we wanted to have a fast account on this first step, hence the Brief Communication format. Please see also our responses to the comments of reviewer #2, we have added a few sentences regarding limitations of this preliminary study.

3. In Fig 2 it is unclear which exact conditions were used in each of the 3/7/11.7 T measurements. Would they be expected to only vary in resolution (i.e. were TE/TR adjusted for the different field T2^(*)/T1) or were they performed under the same conditions (TE/TR)? If the latter, contrast is expected to vary, and to make a good comparison it is recommended to match the contrast conditions while only varying resolution if possible. Please clarify.

This is a very important point, indeed, and inter field comparisons are always difficult, as multiple parameters are at play. The referee is right to wonder about TE. Increasing TE at 3T would increase the contrast, but it would decrease the SNR (exponential decay)

which could be perceived equally unfair. Decreasing the TE would increase the SNR but then decrease the contrast. The results could therefore be biased towards one versus the other. Our goal was to show the change in SNR and CNR quality obtained at 3, 7 and 11.7T using the same set-up (sequence parameters, especially spatial resolution and acquisition time). The parameters therefore were the same: matrix, bandwidth, TR, TE, acquisition time, acceleration etc... TR was fixed to keep the same acquisition time. We are sorry the caption of the figure was not clear, more details about the acquisition parameters at 3 and 7T have been added in the Supplementary Materials section. In the revision, we have added images (Fig.2) acquired with somewhat optimized contrast to noise ratio at 3T and 7T, while keeping the contrast type and acquisition time similar to the 11.7T images, showing that the spatial resolution must necessarily be lower at lower field. But one has to accept that this comparison is arguably not as fair as the SNR comparison given that acquisition parameters (flip angle, TE, etc) were adjusted to compensate for the changes in T1, T2 and T2* values with field strength while not necessarily optimizing the SNR. The main, practical message is that at 11.7T we get high contrast and high resolution while keeping acquisition time short, which is not possible at lower field strength. Please see also our responses to the comments of Reviewer 2.

4. Line 272 TE = 301 ms is correct??

It is correct. For a Space T2-weighted sequence, TE denotes the time at which the center of 2D k-space is acquired.

Rebuttal letter 2:

[Unpublished data redacted]

Version 3:

Decision Letter:

Our ref: NMETH-BC55073C

27th Jun 2024

Dear Dr. Le Bihan,

Thank you for submitting your revised manuscript "First in vivo images of the human brain revealed with the Iseult 11.7T MRI scanner" (NMETH-BC55073C). It has now been seen by the original referees and their comments are below. The reviewers find that the paper has improved in revision, and therefore we'll be happy in principle to publish it in Nature Methods, pending minor revisions to satisfy the referees' final requests and to comply with our editorial and formatting guidelines.

TRANSPARENT PEER REVIEW

ORCID

IMPORTANT: Non-corresponding authors do not have to link their ORCID but are encouraged to do so. Please note that it will not be possible to add/modify ORCIDs at proof. Thus, please let your co-authors know that if they wish to have their ORCID added to the paper they must follow the procedure described in the following link prior to acceptance: <https://www.springernature.com/gp/researchers/orcid/orcid-for-nature-research>

Best regards,
Nina

Nina Vogt, PhD
Senior Editor
Nature Methods

Reviewer #1 (Remarks to the Author):

This is great, thank you.
I have no further comments.

Reviewer #2 (Remarks to the Author):

The authors have answered my major points and clarified where necessary.
I recommend publication of this significant milestone and exciting work.

A couple of comments regarding the minor points:

- I suggest leaving the references to the ISMRM proceedings in the manuscript as they are relevant
- For me it was not clear that Iseult (Isolde) is a musical reference. So, if the scanner is named Iseult, who would then be Tristan?

Reviewer #3 (Remarks to the Author):

I have nothing further to add. Congratulations for this landmark work! Noam Shemesh

Author Rebuttal letter:

RESPONSES TO THE REVIEWERS COMMENTS

Reviewer #1 (Remarks to the Author):

This is great, thank you.
I have no further comments.

Thank you

Reviewer #2 (Remarks to the Author):

The authors have answered my major points and clarified where necessary.
I recommend publication of this significant milestone and exciting work.

A couple of comments regarding the minor points:

- I suggest leaving the references to the ISMRM proceedings in the manuscript as they are relevant
We agree, hence we have kept those references.
- For me it was not clear that Iseult (Isolde) is a musical reference. So, if the scanner is named Iseult, who would then be Tristan?

Yes, Iseult (Isolde) needs Tristan... In fact, Tristan is the name of another project I proposed a few years ago, aiming at conceiving and building a large bore MRI magnet (so called "social magnet") to simultaneously acquire data with fMRI of several subjects interacting. This (serious) project has been presented to CERN and CEA, and is mentioned in my last book *Einstein's Error* (Odile Jacob Publishing, https://www.odilejacob.com/catalogue/sciences/neurosciences/einstein-s-error_9782415001650.php).

There remains an empty arch at NeuroSpin, next to the arch where Iseult is waiting, to house this "social MRI" system, pending appropriate funding and dedicated resources, as the priority is for the moment to fully develop and

exploit Iseult. Perhaps another Nature Methods article in the distant future.
Reviewer #3 (Remarks to the Author):

I have nothing further to add. Congratulations for this landmark work! Noam Shemesh.

Thank you

Version 4:

Decision Letter:

17th Sep 2024

Dear Professor Le Bihan,

I am pleased to inform you that your Brief Communication, "First in vivo images of the human brain revealed with the Iseult 11.7T MRI scanner", has now been accepted for publication in Nature Methods. The received and accepted dates will be February 5th, 2024 and September 17th, 2024. This note is intended to let you know what to expect from us over the next month or so, and to let you know where to address any further questions.

Over the next few weeks, your paper will be copyedited to ensure that it conforms to Nature Methods style. Once your paper is typeset, you will receive an email with a link to choose the appropriate publishing options for your paper and our Author Services team will be in touch regarding any additional information that may be required.

Once proofs are generated, they will be sent to you electronically and you will be asked to send a corrected version within 48 hours. It is extremely important that you let us know now whether you will be difficult to contact over the next month. If this is the case, we ask that you send us the contact information (email, phone and fax) of someone who will be able to check the proofs and deal with any last-minute problems.

If, when you receive your proof, you cannot meet the deadline, please inform us at rjsproduction@springernature.com immediately.

Please note that *Nature Methods* is a Transformative Journal (TJ). Authors may publish their research with us through the traditional subscription access route or make their paper immediately open access through payment of an article-processing charge (APC). Authors will not be required to make a final decision about access to their article until it has been accepted. [Find out more about Transformative Journals](https://www.springernature.com/gp/open-research/transformative-journals)

To assist our authors in disseminating their research to the broader community, our SharedIt initiative provides you with a unique shareable link that will allow anyone (with or without a subscription) to read the published article. Recipients of the link with a subscription will also be able to download and print the PDF. As soon as your article is published, you will receive an automated email with your shareable link.

Please note that you and your coauthors may order reprints and single copies of the issue containing your article through Springer Nature Limited's reprint website, which is located at <http://www.nature.com/reprints/author-reprints.html>. If there are

any questions about reprints please send an email to author-reprints@nature.com and someone will assist you.

Best regards,
Nina

Nina Vogt, PhD
Senior Editor
Nature Methods

** Visit the Springer Nature Editorial and Publishing website at http://www.springernature.com/editorial-and-publishing-jobs?utm_source=ejP_NMeth_email&utm_medium=ejP_NMeth_email&utm_campaign=ejp_Nmeth for more information about our career opportunities. If you have any questions please click [here](mailto:editorial.publishing.jobs@springernature.com).

Open Access This Peer Review File is licensed under a Creative Commons Attribution 4.0 International License, which permits use, sharing, adaptation, distribution and reproduction in any medium or format, as long as you give appropriate credit to the original author(s) and the source, provide a link to the Creative Commons license, and indicate if changes were made. In cases where reviewers are anonymous, credit should be given to 'Anonymous Referee' and the source.
